# Improved-Odd-Even-Prime Reconfiguration to Enhance the Output Power of Rectangular Photovoltaic Array under Partial Shading Conditions

**Dileep Katiki [1], Chandrasekhar Yammani [2] and Surender Reddy Salkuti [3,*]**

[1] Department of Electrical and Electronics Engineering, National Institute of Technology Goa, Ponda 403401, Goa, India
[2] Department of Electrical Engineering, National Institute of Technology, Warangal 506004, Telangana, India
[3] Department of Railroad and Electrical Engineering, Woosong University, Daejeon 34606, Republic of Korea
* Correspondence: surender@wsu.ac.kr

**Abstract:** The output power of a PV (photovoltaic) array decreases due to the partial shading condition (PSC). If one or more PV modules receives less irradiance compared to other PV modules in a PV array then it is called PSC. To enhance the output power of PV array under PSC, Sudoku, and Optimal Sudoku Reconfiguration techniques are available in the literature for perfect squared PV arrays, such as $4 \times 4$, $9 \times 9$, and $16 \times 16$. Odd–Even and Odd–Even–Prime reconfigurations are available for rectangular PV arrays. To enhance the output power of rectangular PV array further, this paper proposed a new reconfiguration technique called Improved Odd–Even–Prime reconfiguration. To validate the proposed method, two PV arrays $9 \times 9$ and $8 \times 9$ have been implemented.

**Keywords:** PV array modeling; static reconfiguration; Improved Odd–Even–Prime; Odd–Even–Prime; partial shading condition





## 1. Introduction

There are two main reasons for the usage of renewable energy sources (RES). The first is the shortage of conventional energy sources, and the second is the demand for electrical energy [1,2]. Solar energy is the best alternative for conventional energy sources among all RES due to its abundant availability and good response from the consumers [3]. Solar PV (photovoltaic) systems are a popular source of renewable energy that use solar panels to convert sunlight into electricity. These systems are composed of a number of components, including solar panels, inverters, and a mounting system. Solar panels are made up of photovoltaic cells that are responsible for converting sunlight into electricity. When sunlight hits a solar panel, it causes the electrons in the photovoltaic cells to become excited and move around. This movement of electrons creates an electric current, which can be used to power homes, businesses, and other structures. Solar PV systems have a number of benefits, including the ability to generate clean, renewable energy and the potential to reduce electricity costs. They also have a low impact on the environment, as they do not produce greenhouse gases or other pollutants. Solar PV systems are a reliable and cost-effective source of renewable energy that can help to reduce our reliance on fossil fuels and protect the environment.

The output power of a PV array is affected by environmental factors such as insulation, temperature, and PSC [4,5]. Among all the factors, PSC reduces the output power of the PV array significantly. Photovoltaic (PV) reconfiguration is a crucial technique for maintaining the efficiency and productivity of PV systems when they are subjected to partial shading conditions. Partial shading occurs when some of the PV modules in a PV array are obstructed from direct sunlight by objects, such as trees, buildings, or other structures. This can significantly decrease the power output of the PV system and may

even cause it to stop functioning completely. PSC occurs if some PV modules receive less irradiance due to the shadow of nearby trees, nearby buildings, and passing clouds, etc. Partially shaded PV modules operate in reverse bias, i.e., they consume power instead of generating power. Hence, temperature of partially shaded PV modules increases and creates a hot-spot problem. Diodes are connected across the PV modules in an anti-parallel fashion to overcome the hot-spot problem. These diodes are called bypass diodes. Several peaks in the I-V characteristics of the PV array are primarily caused by bypass diodes, hence traditional MPPT algorithms may trap at local minima. To overcome the above problem, most of the researchers used nature inspired optimization methods to trace the MPPT in PV arrays [6–10]. PV reconfiguration involves altering the electrical connections between PV modules in order to redirect the flow of current and voltage in a way that allows the PV system to continue operating at optimal levels. By carefully designing the PV reconfiguration strategy, it is possible to significantly reduce the negative impact of partial shading on the power output of the PV system.

The performance of PV array under PSC relies on the shading area, position of shading, and PV array configuration [11–14]. Among all the configurations, TCT having superior performance under PSC [15,16]. Even though TCT configuration has better performance under PSC, the mismatched power loss will increase significantly if more PV modules in a row are subject to a partial-shading condition. Thus, the efficiency of the PV array decreases [17]. To solve the above problem, most of the researchers implemented the "Reconfiguration Technique" [18]. There are various methods of PV reconfiguration, each with its own set of benefits and drawbacks. The most suitable approach will depend on the specific characteristics of the PV system and the shading conditions it faces. For example, some PV reconfiguration techniques are only suitable for use with certain types of PV arrays or under specific shading conditions. In literature, static reconfiguration [19] and dynamic reconfiguration [20] techniques are available. PV reconfiguration is vital because it enables PV systems to continue operating at high levels of efficiency and productivity under partial-shading conditions, which would otherwise significantly reduce their power output. By implementing appropriate PV reconfiguration strategies, it is possible to significantly improve the performance and reliability of PV systems.

In the static reconfiguration technique [21], the position of PV modules is changed, but electrical connections are unaltered. Sudoku PV reconfiguration is a technique used to improve the power output of PV systems under partial-shading conditions. Sudoku PV reconfiguration involves changing the electrical connections between PV modules in a specific pattern based on the Sudoku puzzle game. By carefully designing the Sudoku reconfiguration strategy, it is possible to significantly reduce the negative impact of partial-shading on the power output of the PV system. In the dynamic reconfiguration technique [22,23], the electrical connections of PV modules are changed dynamically with the help of sensors, switches, and control algorithms based on the irradiance of the PV modules. In [24], to supply the maximum power to the motor, a control algorithm for electrical array reconfiguration has been implemented using fuzzy logic controller. In [25], adaptive PV reconfiguration has been implemented to mitigate the effect of PSC. Adaptive PV reconfiguration consists of a fixed part, an adaptive part, and a control algorithm. Based on the control algorithm, adaptive parts of the PV cell are connected to a fixed part of PV cells to reduce the PSC effect. In [26], electrical array reconfiguration (EAR) has been implemented to reduce the power loss due to PSC. In EAR, electrical connection changes according to the controllable switching matrix. The dynamic reconfiguration technique requires a greater number of sensors and switches [27]. The implementation of dynamic reconfiguration is more costly, as well as complex, due to its control algorithms [28,29]. Various papers [30,31] have implemented a dynamic reconfiguration technique to improve the output power under PSC.

There are several advantages to using Sudoku PV reconfiguration. One of the main advantages is that it is relatively simple to implement and does not require any additional hardware or equipment. However, there are also some limitations to using Sudoku PV

reconfiguration. One limitation is that it is only effective for PV arrays with a perfect square shape, such as 4 × 4, 9 × 9, and 16 × 16 arrays. It is not suitable for PV arrays with other shapes or sizes. It is simple to implement.

In papers [32,33], Sudoku and its optimal version of PV static reconfiguration techniques are applied to improve the output power under partial-shading conditions. These methods are only appropriate for PV arrays that are perfectly squared, such as 4 × 4, 9 × 9, and 16 × 16. In [34], a new static reconfiguration approach based on a screw pattern in row formation is proposed. This screw-propagated array configuration is further divided into two categories based on the direction of propagation: horizontal and vertical. The Ken-Ken puzzle-based reconfiguration has been implemented in [35] to enhance the output power under partial-shading conditions. The results are compared with Sudoku reconfiguration. In [36], the authors implemented a static reconfiguration based on the moves of the Knight piece in a chess game. This reconfiguration works for all squared PV arrays. In [37,38], authors proposed one-time 'Odd–Even' and 'Odd–Even–Prime' (OEP) reconfigurations to enhance the output power of rectangular PV arrays under partial-shading conditions. To enhance the output power of rectangular PV array further, this paper proposed a new reconfiguration technique called "Improved Odd-Even-Prime (IOEP)" reconfiguration. To validate the proposed method two PV arrays 9 × 9 and 8 × 9 haven implemented. The performance of the proposed technique studied with global maximum power point (GMPP), mismatch power loss (MPL), fill-factor (FF), and efficiency indices.

*Novelty of the Paper*

The novelty of this paper explained below as:

- TCT configuration generates more power for column shading compared to row shading.
- Therefore, to enhance the output power under PSC, one should convert row shading into column shading.
- A reconfiguration technique used to convert row shading into column shading.
- For squared PV arrays, conversion of row shading into column shading is possible, but for rectangular PV arrays it is not possible perfectly, so a maximum portion of row shading should be converted to column shading to generate more power. To convert a maximum portion of row shading into column shading, the same numbered rows should not appear.
- In OEP reconfiguration, same-numbered rows appeared adjacently. To overcome the above problem, IOEP reconfiguration has been introduced. In IOEP reconfiguration, same-numbered rows do not appear adjacently. Hence the performance of IOEP reconfiguration is better than the OEP reconfiguration.

In the 9 × 9 OEP arrangement, in any row maximum of Four PV modules which belongs to the same row are presented, as shown in Figure 1a, whereas in a 9 × 9 IOEP arrangement, in any row maximum of three PV modules which belongs to the same row are presented as shown in Figure 1b. In contrast to OEP reconfiguration, IOEP reconfiguration effectively enables shade dispersion to all feasible rows.

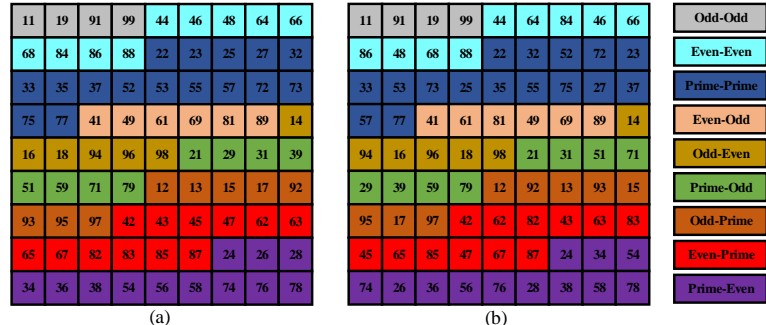

**Figure 1.** (**a**) OEP and (**b**) IOEP arrangement for 9 × 9 PV array.

The rest of manuscript is organised as follows: Section 2, formation of IOEP and OEP reconfigurations. In Section 3, the results for IOEP and OEP under various partial-shading conditions are summarized, followed by the conclusion, which is presented in Section 4.

## 2. Improved-Odd-Even-Prime

In OEP reconfiguration [38], the PV modules with same numbered rows are placed adjacently ($M_{11}$, $M_{19}$, $M_{91}$, $M_{99}$); due to this arrangement the shade will not distribute properly. To overcome the above problem, this paper introduced IOEP reconfiguration; in IOEP reconfiguration (IOEP), the PV modules with same numbered rows are not placed adjacently ($M_{11}$, $M_{91}$, $M_{19}$, $M_{99}$) as much as possible. Due to this type of arrangement, the shade will distribute properly compared to OEP reconfiguration, hence the output power of the PV array increased compared to OEP reconfiguration.

Consider a PV array of size $m \times n$ for example, where m stands for the number of rows and n for the number of columns. Each PV module is identified by the letters $M_{xy}$, where M is the number of the module, and x (x = 1, 2, 3, . . . m) and y (y = 1, 2, 3, . . . n) are the row and column numbers, respectively. For instance, the module in the fourth row and seventh column of the TCT configuration is referred to as $M_{47}$.

For the purpose of analyzing the suggested "IOEP" layout, PV array of orders $8 \times 9$ and $9 \times 9$ are chosen. The PV array rows and columns are separated into three groups: prime numbers, odd numbers without primes, and even numbers without primes.

Modules are arranged in the following sequence: odd-numbered rows and columns, even-numbered rows and columns, prime-numbered rows and columns, even–odd, odd–even, prime–odd, odd–prime, even–prime, and prime–even. In this arrangement, the modules in odd-numbered rows and columns are placed first (e.g., $M_{11}$ $M_{91}$, $M_{19}$ $M_{99}$. . .), followed by those in even-numbered rows and columns (e.g., $M_{22}$ $M_{32}$, $M_{52}$), and so on.

The PV array, which has a size of $m \times n$, can be divided into three groups.

To obtain the "IOEP" arrangement follow the below steps:

Step 1: First place ONR and ONC.

For $9 \times 9$ PV array, the odd-numbered rows = [1,9] and the odd-numbered columns = [1 9], then

Odd–odd numbers are $\begin{bmatrix} 11 & 19 \\ 91 & 99 \end{bmatrix}$

OEP arrangement: ($M_{11}$, $M_{19}$, $M_{91}$, $M_{99}$)

Improved-OEP arrangement: ($M_{11}$, $M_{91}$, $M_{19}$, $M_{99}$)

Step 2: Place ENR and ENC. For $9 \times 9$ PV array, the even-numbered rows = [4 6 8] and the even-numbered columns = [4 6 8], then

Even–even numbers are $\begin{bmatrix} 44 & 46 & 48 \\ 64 & 66 & 68 \\ 84 & 86 & 88 \end{bmatrix}$

OEP arrangement: ($M_{44}$, $M_{46}$, $M_{48}$, $M_{64}$, $M_{66}$, $M_{68}$, $M_{84}$, $M_{86}$, $M_{88}$)

Improved-OEP arrangement: ($M_{44}$, $M_{64}$, $M_{84}$, $M_{46}$, $M_{66}$, $M_{86}$, $M_{48}$, $M_{68}$, $M_{88}$)

Step 3: Place PNR and PNC.

For $9 \times 9$ PV array, the prime-numbered rows = [4 6 8] and the prime-numbered columns = [4 6 8], then

Prime–prime numbers are $\begin{bmatrix} 22 & 23 & 25 & 27 \\ 32 & 33 & 35 & 37 \\ 52 & 53 & 55 & 57 \\ 72 & 73 & 75 & 77 \end{bmatrix}$

OEP arrangement: ($M_{22}$, $M_{23}$, $M_{25}$, $M_{27}$, $M_{32}$, $M_{33}$, $M_{35}$, $M_{37}$, $M_{52}$, . . . , $M_{77}$)

Improved-OEP arrangement: ($M_{22}$, $M_{32}$, $M_{52}$, $M_{72}$, $M_{23}$, $M_{33}$, $M_{53}$, $M_{73}$, $M_{25}$, . . . , $M_{77}$)

Step 4: Place ENR and ONC.

For $9 \times 9$ PV array, the even-numbered rows = [4 6 8] and the odd-numbered columns = [1 9], then

Even–odd numbers are $\begin{bmatrix} 41 & 49 \\ 61 & 69 \\ 81 & 89 \end{bmatrix}$

OEP arrangement: $(M_{41}, M_{49}, M_{61}, M_{69}, M_{81}, M_{89})$

Improved-OEP arrangement: $(M_{41}, M_{61}, M_{81}, M_{49}, M_{69}, M_{89})$

Step 5: Place ONR and ENC

For $9 \times 9$ PV array, the odd-numbered rows = [1 9] and the even-numbered columns = [4 6 8], then

Odd–even numbers are $\begin{bmatrix} 14 & 16 & 18 \\ 94 & 96 & 98 \end{bmatrix}$

OEP arrangement: $(M_{14}, M_{16}, M_{18}, M_{94}, M_{96}, M_{98})$

Improved-OEP arrangement: $(M_{14}, M_{94}, M_{16}, M_{96}, M_{18}, M_{98})$

Step 6: Place PNR and ONC

For $9 \times 9$ PV array, the prime-numbered rows = [2 3 5 7] and the odd-numbered columns = [1 9], then

Prime–odd numbers are $\begin{bmatrix} 21 & 29 \\ 31 & 39 \\ 51 & 59 \\ 71 & 79 \end{bmatrix}$

OEP arrangement: $(M_{21}, M_{29}, M_{31}, M_{39}, M_{51}, M_{59}, M_{71}, M_{79})$

Improved-OEP arrangement: $(M_{21}, M_{31}, M_{51}, M_{71}, M_{29}, M_{39}, M_{59}, M_{79})$

Step 7: Place ONR and PNC

For $9 \times 9$ PV array, the odd-numbered rows = [1 9] and the prime-numbered columns = [2 3 5 7], then

Odd–prime numbers are $\begin{bmatrix} 12 & 13 & 15 & 17 \\ 92 & 93 & 95 & 97 \end{bmatrix}$

OEP arrangement: $(M_{21}, M_{13}, M_{15}, M_{17}, M_{92}, M_{93}, M_{95}, M_{97})$

Improved-OEP arrangement: $(M_{12}, M_{92}, M_{13}, M_{93}, M_{15}, M_{95}, M_{17}, M_{97})$

Step 8: Place ENR and PNC

For $9 \times 9$ PV array, the even-numbered rows = [4 6 8] and the prime-numbered columns = [2 3 5 7], then

Even–prime numbers are $\begin{bmatrix} 42 & 43 & 45 & 47 \\ 62 & 63 & 65 & 67 \\ 82 & 83 & 85 & 87 \end{bmatrix}$

OEP arrangement: $(M_{42}, M_{43}, M_{45}, M_{47}, M_{62}, M_{63}, M_{65}, M_{67}, M_{82}, \ldots, M_{87})$

Improved-OEP arrangement: $(M_{42}, M_{62}, M_{82}, M_{43}, M_{63}, M_{83}, M_{45}, M_{65}, M_{85}, M_{47}, \ldots, M_{87})$

Step 9: Place PNR and ENC

For $9 \times 9$ PV array, the prime-numbered rows = [2 3 5 7] and the even-numbered columns = [4 6 8], then

Even–prime numbers are $\begin{bmatrix} 24 & 26 & 28 \\ 34 & 36 & 38 \\ 54 & 53 & 58 \\ 74 & 76 & 78 \end{bmatrix}$

OEP arrangement: $(M_{24}, M_{26}, M_{28}, M_{34}, M_{36}, M_{38}, M_{54}, M_{56}, M_{58}, \ldots, M_{78})$

Improved-OEP arrangement: $(M_{24}, M_{34}, M_{54}, M_{74}, M_{26}, M_{36}, M_{56}, M_{76}, M_{28}, M_{38}, \ldots, M_{78})$

Figure 1 illustrates the OEP and IOEP arrangements for a $9 \times 9$ TCT-configured PV array. In the $9 \times 9$ OEP arrangement, a maximum of four PV modules belonging to the same row are displayed in each row, as shown in Figure 1a. In the $9 \times 9$ IOEP arrangement, a maximum of three PV modules belonging to the same row are displayed in each row, as

shown in Figure 1b. Therefore, the IOEP reconfiguration allows for more effective shade dispersion across all rows compared to the OEP reconfiguration.

Three performance indices, fill-factor (FF), mismatch power loss (MPL), and efficiency are used to gauge how well the IOEP setup performs in partial shade situations. Fill-factor (FF) is a measure of the efficiency of a photovoltaic (PV) cell or module. It is defined as the ratio of the maximum power that a PV cell or module can produce to the product of its open-circuit voltage (Voc) and short-circuit current (Isc). FF is an important parameter in PV cells and modules because it determines the amount of power that can be generated under real-world operating conditions. A PV cell or module with a high FF is able to generate more power than one with a low FF, even if they have the same Voc and Isc. There are several factors that can affect the FF of a PV cell or module, including the quality and purity of the materials used, the thickness and surface area of the active layers, and the efficiency of the charge carriers. In general, higher FF values are desirable in PV cells and modules because they result in higher power output and efficiency. FF is an important measure of the efficiency and performance of PV cells and modules, and it is important to consider when designing and evaluating PV systems.

Mismatch power loss (MPL) refers to the loss of energy that occurs when the power output of the photovoltaic (PV) cells in a PV system do not match up with one another. This can occur when the cells are not evenly shaded, when there is a difference in their efficiency, or when they are operating at different temperatures. MPL can significantly impact the overall efficiency of a PV system, reducing its ability to generate electricity. It can also lead to a reduction in the lifespan of the PV cells, as the cells that are producing less power are subjected to higher levels of stress. There are a few ways to mitigate MPL in PV systems. One is to carefully design the system to ensure that the PV cells are evenly shaded and receive an equal amount of sunlight. This can be achieved through the use of shading devices or by carefully positioning the cells, in our case reconfiguration. MPL is an important consideration in the design and operation of PV systems, as it can significantly impact the efficiency and lifespan of the system. By understanding and addressing the factors that contribute to MPL, it is possible to optimize the performance of a PV system and maximize its ability to generate electricity.

Efficiency in photovoltaic (PV) systems refers to the ability of the system to convert sunlight into electricity. The efficiency of a PV system is typically expressed as a percentage, and it is calculated by dividing the amount of electricity that the system generates by the amount of sunlight that it receives. There are a number of factors that can impact the efficiency of a PV system. One of the most important is reducing the PSC. Another factor that can impact the efficiency of a PV system is the temperature at which the cells are operating. The orientation and angle of the PV cells can also affect the efficiency of a system. In general, PV cells that are positioned to receive direct sunlight will be more efficient than those that are shaded or receive diffuse light. Finally, the overall design of the PV system can impact its efficiency. Carefully designing the system to minimize losses due to factors such as mismatch power loss and resistance can help to improve its efficiency. The efficiency of a PV system is an important consideration, as it determines the amount of electricity that the system is able to generate. By understanding and addressing the factors that impact efficiency, it is possible to optimize the performance of a PV system and maximize its ability to generate electricity.

### 3. Results and Discussion

In order to increase the output power of PV arrays under various partial shadowing conditions, the proposed "IOEP" technique has been used in this article to $9 \times 9$ and $8 \times 9$ TCT layouts. These shading conditions were taken from the OEP study [38]. The maximum output power values were calculated theoretically for each partial shading condition and verified using MATLAB/SIMULINK. The parameters for PV modules that were taken into consideration are listed in Table 1. Performance for the new "IOEP" arrangement and the existing "OEP" [38] layout were compared.

**Table 1.** The Parameters of PV module.

| Parameter | Rating |
|---|---|
| Rated Power | 170 W |
| Open-Circuit Voltage $V_{oc}$ | 44.2 V |
| Short-Circuit Current $I_{sc}$ | 5.2 A |
| Current at MPP $I_{mp}$ | 4.75 A |
| Voltage at MPP $V_{mp}$ | 35.8 A |
| Number of Cells | 72 |
| PV Module Area | 62.2 inc $\times$ 31.9 inc |

For accurate modeling of PV module forward–backward linear least-square error method used [39].

### 3.1. Implementation of Improved Odd–Even–Prime for 9 × 9 PV Array

3.1.1. Case 1 (Top-Left) Shading

In Case-1, a 4 × 4 sub array subjected to PSC at Top-Left as shown in Figure 2a.

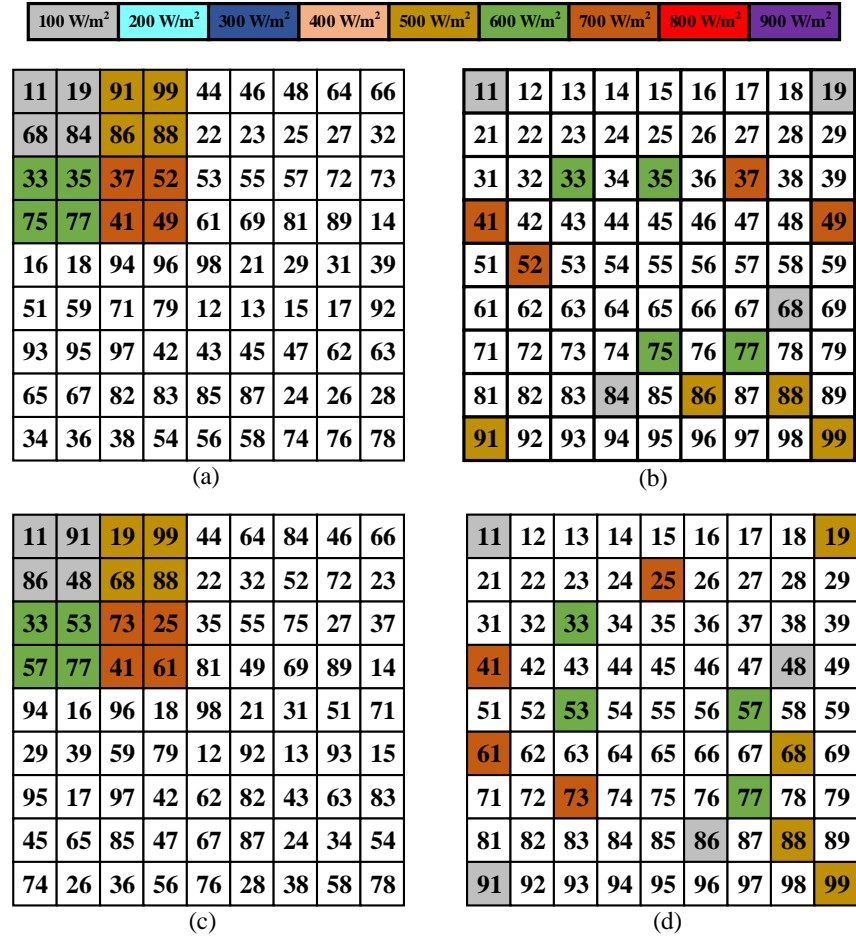

**Figure 2.** (**a**) OEP (**b**) Shade dispersion of OEP (**c**) IOEP (**d**) Shade dispersion of Improved Odd–Even–Prime for 9 × 9 Top-Left PSC.

### 3.1.2. Calculation of GMPP for OEP

The shade distribution of OEP is depicted in Figure 2b. The row currents are

$$\left.\begin{array}{lll} I_{row-1} = 7.2 I_m & I_{row-2} = 9 I_m & I_{row-3} = 7.9 I_m \\ I_{row-4} = 8.4 I_m & I_{row-5} = 8.7 I_m & I_{row-6} = 8.1 I_m \\ I_{row-7} = 8.2 I_m & I_{row-8} = 7.1 I_m & I_{row-9} = 8 I_m \end{array}\right\} \quad (1)$$

If the PV array current is 7.2 $I_m$ then row-8 will bypass, hence the PV array voltage is 8 $V_m$ and the PV array power is 57.6 $V_m I_m$. If the PV array 7.1 $I_m$ then all the rows of PV array can carry this current hence no row will bypass, therefore PV array voltage is 9 $V_m$ and PV array power is 63.9 $V_m I_m$. For different values of array current, the array voltage and power are calculated and noted in Table 1.

### 3.1.3. Calculation of GMPP for Improved-OEP

Figure 2d depicts the IOEP's shade dispersion. The estimated row currents

$$\left.\begin{array}{lll} I_{row-1} = 7.6 I_m & I_{row-2} = 8.7 I_m & I_{row-3} = 8.6 I_m \\ I_{row-4} = 7.8 I_m & I_{row-5} = 8.2 I_m & I_{row-6} = 8.2 I_m \\ I_{row-7} = 8.3 I_m & I_{row-8} = 7.6 I_m & I_{row-9} = 7.6 I_m \end{array}\right\} \quad (2)$$

For different values of array current, the array voltage and power are calculated and noted in Table 2. From Table 2, it is clearly understood that the maximum value of power for OEP is **63.9** $V_m I_m$ and the maximum value of power for IOEP is **68.4** $V_m I_m$.

**Table 2.** GMPP for $9 \times 9$ PV array for Case-1 (Top-Left) PSC.

| Odd-Even-Primve | | | | Improved-Odd-Even-Primve | | | |
|---|---|---|---|---|---|---|---|
| **Row Bypassed** | $I_a$ | $V_a$ | $P_a$ | **Row Bypassed** | $I_a$ | $V_a$ | $P_a$ |
| $I_{R9}$ | 8.0 $I_m$ | 6 $V_m$ | 48.0 $V_m I_m$ | $I_{R9}$ | 7.6 $I_m$ | 9 $V_m$ | 68.4 $V_m I_m$ |
| $I_{R8}$ | 7.1 $I_m$ | 9 $V_m$ | **63.9** $V_m I_m$ | $I_{R8}$ | 7.6 $I_m$ | 9 $V_m$ | **68.4** $V_m I_m$ |
| $I_{R7}$ | 8.2 $I_m$ | 4 $V_m$ | 32.8 $V_m I_m$ | $I_{R7}$ | 8.3 $I_m$ | 3 $V_m$ | 24.9 $V_m I_m$ |
| $I_{R6}$ | 8.1 $I_m$ | 5 $V_m$ | 40.5 $V_m I_m$ | $I_{R6}$ | 8.2 $I_m$ | 5 $V_m$ | 41 $V_m I_m$ |
| $I_{R5}$ | 8.7 $I_m$ | 2 $V_m$ | 17.4 $V_m I_m$ | $I_{R5}$ | 8.2 $I_m$ | 5 $V_m$ | 41 $V_m I_m$ |
| $I_{R4}$ | 8.4 $I_m$ | 3 $V_m$ | 25.2 $V_m I_m$ | $I_{R4}$ | 7.8 $I_m$ | 6 $V_m$ | 46.8 $V_m I_m$ |
| $I_{R3}$ | 7.9 $I_m$ | 7 $V_m$ | 55.3 $V_m I_m$ | $I_{R3}$ | 8.6 $I_m$ | 2 $V_m$ | 17.2 $V_m I_m$ |
| $I_{R2}$ | 9.0 $I_m$ | 1 $V_m$ | 9.0 $V_m I_m$ | $I_{R2}$ | 8.7 $I_m$ | 1 $V_m$ | 8.7 $V_m I_m$ |
| $I_{R1}$ | 7.2 $I_m$ | 8 $V_m$ | 57.6 $V_m I_m$ | $I_{R1}$ | 7.6 $I_m$ | 9 $V_m$ | 68.4 $V_m I_m$ |

The voltage–power characteristics for $9 \times 9$ PV array for case-1 (Top-Left) are shown in the Figure 3. From the voltage–power characteristics it understood that IOEP reconfiguration having superior performance under partial-shading conditions.

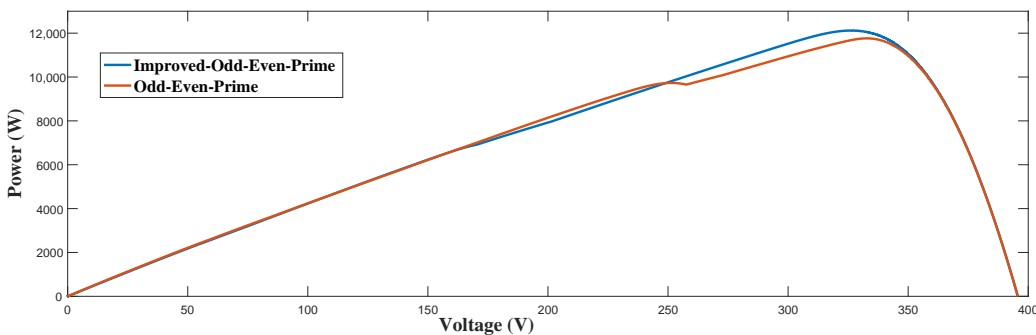

**Figure 3.** Voltage–power characteristics of 9 × 9 PV array for case-1 (Top-Left) PSC.

### 3.1.4. Case-2 (Top-Right) Shading

In Case-2, a 4 × 4 sub-array is subjected to partial shading condition at Top-Right as shown in Figure 4a.

| 100 W/m² | 200 W/m² | 300 W/m² | 400 W/m² | 500 W/m² | 600 W/m² | 700 W/m² | 800 W/m² | 900 W/m² |
|---|---|---|---|---|---|---|---|---|

| 11 | 19 | 91 | 99 | 44 | 46 | 48 | 64 | 66 |
| 68 | 84 | 86 | 88 | 22 | 23 | 25 | 27 | 32 |
| 33 | 35 | 37 | 52 | 53 | 55 | 57 | 72 | 73 |
| 75 | 77 | 41 | 49 | 61 | 69 | 81 | 89 | 14 |
| 16 | 18 | 94 | 96 | 98 | 21 | 29 | 31 | 39 |
| 51 | 59 | 71 | 79 | 12 | 13 | 15 | 17 | 92 |
| 93 | 95 | 97 | 42 | 43 | 45 | 47 | 62 | 63 |
| 65 | 67 | 82 | 83 | 85 | 87 | 24 | 26 | 28 |
| 34 | 36 | 38 | 54 | 56 | 58 | 74 | 76 | 78 |

(a)

| 11 | 12 | 13 | 14 | 15 | 16 | 17 | 18 | 19 |
| 21 | 22 | 23 | 24 | 25 | 26 | 27 | 28 | 29 |
| 31 | 32 | 33 | 34 | 35 | 36 | 37 | 38 | 39 |
| 41 | 42 | 43 | 44 | 45 | 46 | 47 | 48 | 49 |
| 51 | 52 | 53 | 54 | 55 | 56 | 57 | 58 | 59 |
| 61 | 62 | 63 | 64 | 65 | 66 | 67 | 68 | 69 |
| 71 | 72 | 73 | 74 | 75 | 76 | 77 | 78 | 79 |
| 81 | 82 | 83 | 84 | 85 | 86 | 87 | 88 | 89 |
| 91 | 92 | 93 | 94 | 95 | 96 | 97 | 98 | 99 |

(b)

| 11 | 91 | 19 | 99 | 44 | 64 | 84 | 46 | 66 |
| 86 | 48 | 68 | 88 | 22 | 32 | 52 | 72 | 23 |
| 33 | 53 | 73 | 25 | 35 | 55 | 75 | 27 | 37 |
| 57 | 77 | 41 | 61 | 81 | 49 | 69 | 89 | 14 |
| 94 | 16 | 96 | 18 | 98 | 21 | 31 | 51 | 71 |
| 29 | 39 | 59 | 79 | 12 | 92 | 13 | 93 | 15 |
| 95 | 17 | 97 | 42 | 62 | 82 | 43 | 63 | 83 |
| 45 | 65 | 85 | 47 | 67 | 87 | 24 | 34 | 54 |
| 74 | 26 | 36 | 56 | 76 | 28 | 38 | 58 | 78 |

(c)

| 11 | 12 | 13 | 14 | 15 | 16 | 17 | 18 | 19 |
| 21 | 22 | 23 | 24 | 25 | 26 | 27 | 28 | 29 |
| 31 | 32 | 33 | 34 | 35 | 36 | 37 | 38 | 39 |
| 41 | 42 | 43 | 44 | 45 | 46 | 47 | 48 | 49 |
| 51 | 52 | 53 | 54 | 55 | 56 | 57 | 58 | 59 |
| 61 | 62 | 63 | 64 | 65 | 66 | 67 | 68 | 69 |
| 71 | 72 | 73 | 74 | 75 | 76 | 77 | 78 | 79 |
| 81 | 82 | 83 | 84 | 85 | 86 | 87 | 88 | 89 |
| 91 | 92 | 93 | 94 | 95 | 96 | 97 | 98 | 99 |

(d)

**Figure 4.** (**a**) OEP; (**b**) Shade dispersion of OEP; (**c**) IOEP; (**d**) Shade dispersion of IOEP for 9 × 9 PV array at Top-Right PSC.

The voltage–power characteristics of case-2 (Top-Right) are shown in the Figure 5.

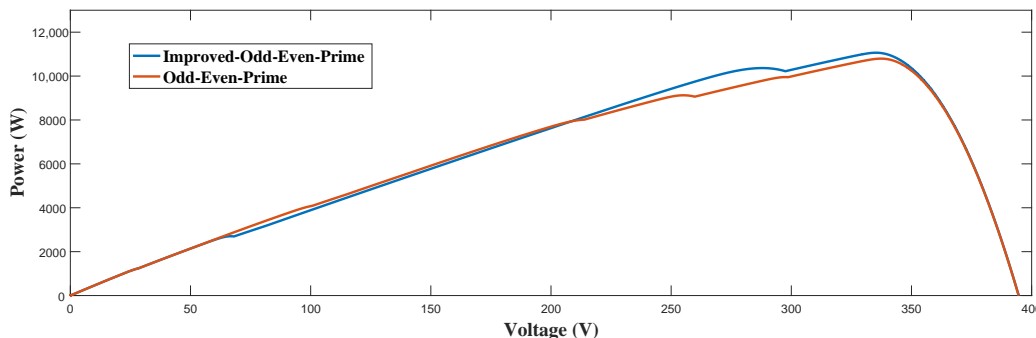

**Figure 5.** Voltage–power characteristics of 9 × 9 PV array for case-2 (Top-Right) PSC.

From the Figure 5, it can be observed that maximum power generated by OEP and IOEP are 10,795 W and 11,064 W, respectively. The power generated by IOEP reconfiguration is 2.5% more than OEP reconfiguration.

For different values of array current, the array voltage and power are calculated and noted in Table 3. From Table 3, it is clearly understood that the maximum value of power for OEP is **57.6** $V_m I_m$ and the maximum value of power for IOEP is **59.4** $V_m I_m$.

**Table 3.** GMPP for 9 × 9 PV array for Case-2 (Top-Right) PSC.

| \multicolumn{4}{c} Odd-Even-Primve | | | | Improved-Odd-Even-Primve | | | |
|---|---|---|---|---|---|---|---|
| Row Bypassed | $I_a$ | $V_a$ | $P_a$ | Row Bypassed | $I_a$ | $V_a$ | $P_a$ |
| $I_{R9}$ | $9.0\ I_m$ | $1\ V_m$ | $9.0\ V_m I_m$ | $I_{R9}$ | $9.0\ I_m$ | $1\ V_m$ | $9 V_m I_m$ |
| $I_{R8}$ | $7.7\ I_m$ | $5\ V_m$ | $38.5\ V_m I_m$ | $I_{R8}$ | $7.5\ I_m$ | $7\ V_m$ | $52.5\ V_m I_m$ |
| $I_{R7}$ | $7.8\ I_m$ | $4\ V_m$ | $31.2\ V_m I_m$ | $I_{R7}$ | $7.5\ I_m$ | $7\ V_m$ | $52.5\ V_m I_m$ |
| $I_{R6}$ | $6.7\ I_m$ | $8\ V_m$ | $53.6\ V_m I_m$ | $I_{R6}$ | $6.6\ I_m$ | $9\ V_m$ | **$59.4\ V_m I_m$** |
| $I_{R5}$ | $7.6\ I_m$ | $6\ V_m$ | $45.6\ V_m I_m$ | $I_{R5}$ | $7.4\ I_m$ | $8\ V_m$ | $59.2\ V_m I_m$ |
| $I_{R4}$ | $7.2\ I_m$ | $7\ V_m$ | $50.4\ V_m I_m$ | $I_{R4}$ | $7.5\ I_m$ | $7\ V_m$ | $52.5\ V_m I_m$ |
| $I_{R3}$ | $8.2\ I_m$ | $3\ V_m$ | $24.6\ V_m I_m$ | $I_{R3}$ | $7.5\ I_m$ | $7\ V_m$ | $52.5\ V_m I_m$ |
| $I_{R2}$ | $6.4\ I_m$ | $9\ V_m$ | **$57.6\ V_m I_m$** | $I_{R2}$ | $7.6\ I_m$ | $3\ V_m$ | $22.8\ V_m I_m$ |
| $I_{R1}$ | $8.4\ I_m$ | $2\ V_m$ | $16.8\ V_m I_m$ | $I_{R1}$ | $8.4\ I_m$ | $2\ V_m$ | $16.8\ V_m I_m$ |

### 3.1.5. Case-3 (Bottom-Left) Shading

In Case-3, In the Bottom-Left corner, 16 PV modules are partially shaded, as seen in the Figure 6a.

The voltage–power characteristics of case-3 (Bottom-Left) are shown in the Figure 7. From the voltage–power characteristics it understood that IOEP reconfiguration having superior performance under partial-shading conditions.

For different values of array current, the array voltage and power are calculated and noted in Table 4. From Table 4, it is clearly understood that the maximum value of power for OEP is **57.6** $V_m I_m$ and the maximum value of power for IOEP is **63** $V_m I_m$.

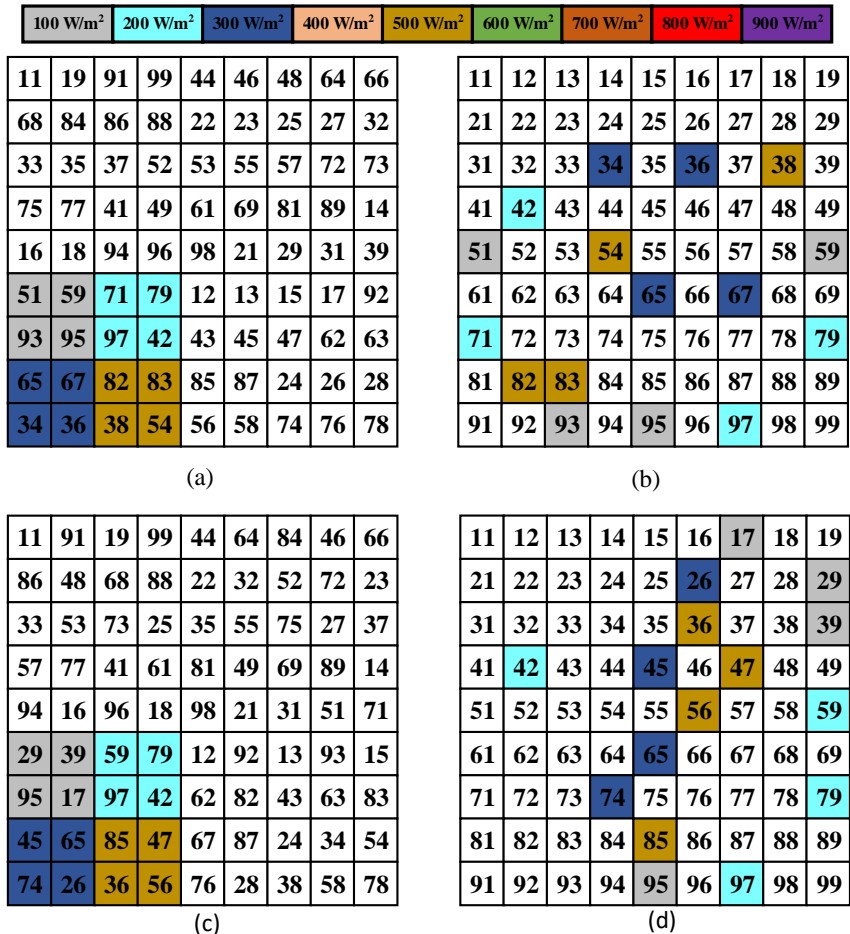

**Figure 6.** (**a**) OEP; (**b**) shade dispersion of OEP; (**c**) IOEP; (**d**) shade dispersion of IOEP for 9 × 9 PV array at Bottom-Left PSC.

### 3.1.6. Case-4 (Bottom-Right) Shading

In Case-IV, in the Bottom-Right corner, 16 PV modules are partially shaded, as seen in the Figure 8a.

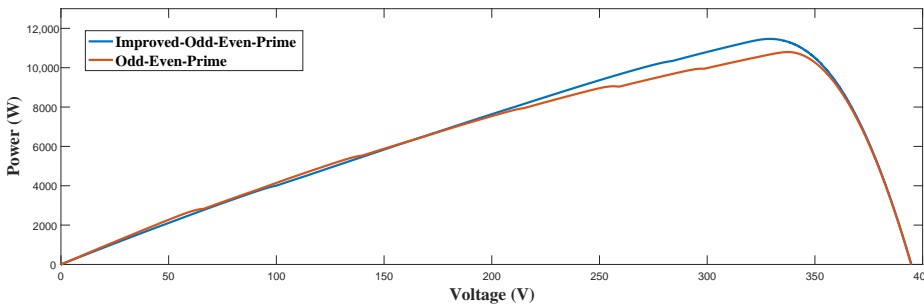

**Figure 7.** Voltage–power characteristics of 9 × 9 PV array for case-3 (Bottom-Left) PSC.

**Table 4.** GMPP for $9 \times 9$ PV array for Case-III (Bottom-Left) PSC.

| | Odd–Even–Prime | | | | Improved Odd–Even–Prime | | |
|---|---|---|---|---|---|---|---|
| Row Bypassed | $I_a$ | $V_a$ | $P_a$ | Row Bypassed | $I_a$ | $V_a$ | $P_a$ |
| $I_{R9}$ | 6.4 $I_m$ | 9 $V_m$ | **57.6** $V_m I_m$ | $I_{R9}$ | 7.3 $I_m$ | 8 $V_m$ | 58.4 $V_m I_m$ |
| $I_{R8}$ | 8.0 $I_m$ | 4 $V_m$ | 32.0 $V_m I_m$ | $I_{R8}$ | 8.5 $I_m$ | 1 $V_m$ | 8.5 $V_m I_m$ |
| $I_{R7}$ | 7.4 $I_m$ | 6 $V_m$ | 44.4 $V_m I_m$ | $I_{R7}$ | 7.5 $I_m$ | 6 $V_m$ | 45 $V_m I_m$ |
| $I_{R6}$ | 7.6 $I_m$ | 5 $V_m$ | 38.0 $V_m I_m$ | $I_{R6}$ | 8.3 $I_m$ | 2 $V_m$ | 16.6 $V_m I_m$ |
| $I_{R5}$ | 6.7 $I_m$ | 8 $V_m$ | 53.6 $V_m I_m$ | $I_{R5}$ | 7.7 $I_m$ | 4 $V_m$ | 30.8 $V_m I_m$ |
| $I_{R4}$ | 8.2 $I_m$ | 3 $V_m$ | 24.6 $V_m I_m$ | $I_{R4}$ | 7.0 $I_m$ | 9 $V_m$ | **63** $V_m I_m$ |
| $I_{R3}$ | 7.1 $I_m$ | 7 $V_m$ | 49.7 $V_m I_m$ | $I_{R3}$ | 7.6 $I_m$ | 5 $V_m$ | 38 $V_m I_m$ |
| $I_{R2}$ | 9.0 $I_m$ | 2 $V_m$ | 18.0 $V_m I_m$ | $I_{R2}$ | 7.4 $I_m$ | 7 $V_m$ | 51.8 $V_m I_m$ |
| $I_{R1}$ | 9.0 $I_m$ | 2 $V_m$ | 18.0 $V_m I_m$ | $I_{R1}$ | 8.1 $I_m$ | 3 $V_m$ | 24.3 $V_m I_m$ |

**Figure 8.** (**a**) OEP; (**b**) shade dispersion of OEP; (**c**) IOEP; and (**d**) shade dispersion of IOEP for $9 \times 9$ PV array at Bottom-Right PSC.

The voltage–power characteristics of case-IV (Bottom-Right) are shown in the Figure 9.

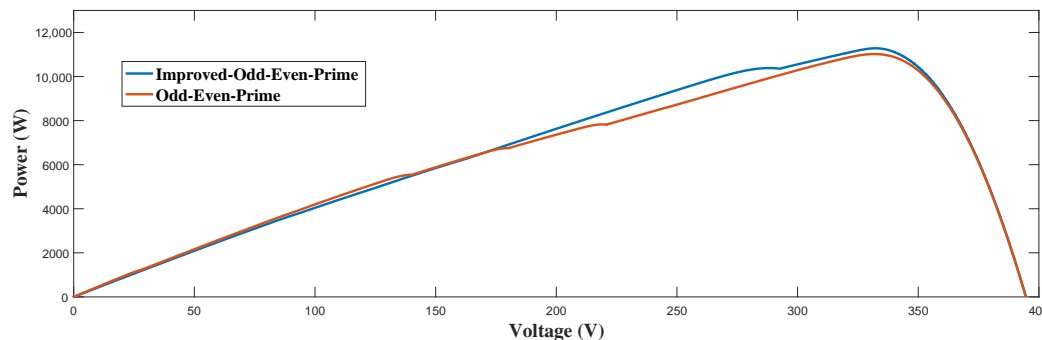

**Figure 9.** Voltage–power characteristics of 9 × 9 PV array for case-4 (Bottom-Right) PSC.

For different values of array current, the array voltage and power are calculated and noted in Table 5. From Table 5, it is clearly understood that the maximum value of power for OEP is **60.3** $V_m I_m$ and the maximum value of power for IOEP is **61.2** $V_m I_m$.

**Table 5.** GMPP for 9 × 9 PV array for Case-4 (Bottom-Right) PSC.

| Odd-Even-Primve | | | | Improved-Odd-Even-Primve | | | |
|---|---|---|---|---|---|---|---|
| Row Bypassed | $I_a$ | $V_a$ | $P_a$ | Row Bypassed | $I_a$ | $V_a$ | $P_a$ |
| $I_{R9}$ | 8.1 $I_m$ | 4 $V_m$ | 32.4 $V_m I_m$ | $I_{R9}$ | 7.4 $I_m$ | 8 $V_m$ | 59.2 $V_m I_m$ |
| $I_{R8}$ | 8.4 $I_m$ | 3 $V_m$ | 25.2 $V_m I_m$ | $I_{R8}$ | 6.8 $I_m$ | 9 $V_m$ | **61.2** $V_m I_m$ |
| $I_{R7}$ | 6.8 $I_m$ | 8 $V_m$ | 54.4 $V_m I_m$ | $I_{R7}$ | 8.2 $I_m$ | 2 $V_m$ | 16.4 $V_m I_m$ |
| $I_{R6}$ | 7.2 $I_m$ | 6 $V_m$ | 43.2 $V_m I_m$ | $I_{R6}$ | 8.1 $I_m$ | 3 $V_m$ | 24.3 $V_m I_m$ |
| $I_{R5}$ | 8.4 $I_m$ | 3 $V_m$ | 25.2 $V_m I_m$ | $I_{R5}$ | 7.4 $I_m$ | 8 $V_m$ | 59.2 $V_m I_m$ |
| $I_{R4}$ | 7.6 $I_m$ | 5 $V_m$ | 38.0 $V_m I_m$ | $I_{R4}$ | 8.3 $I_m$ | 1 $V_m$ | 8.3 $V_m I_m$ |
| $I_{R3}$ | 9.0 $I_m$ | 1 $V_m$ | 9.0 $V_m I_m$ | $I_{R3}$ | 7.6 $I_m$ | 5 $V_m$ | 38 $V_m I_m$ |
| $I_{R2}$ | 6.8 $I_m$ | 8 $V_m$ | 54.4 $V_m I_m$ | $I_{R2}$ | 7.8 $I_m$ | 4 $V_m$ | 31.2 $V_m I_m$ |
| $I_{R1}$ | 6.7 $I_m$ | 9 $V_m$ | **60.3** $V_m I_m$ | $I_{R1}$ | 7.4 $I_m$ | 8 $V_m$ | 59.2 $V_m I_m$ |

### 3.1.7. Case-5 (Top-Left and Bottom-Right) Shading

In case-5, combination of Top-Left and Bottom-Right shadings haven been considered.

In case-5, a total of 16 PV modules were subjected to partial shading condition. The shade dispersion of Top-Left and Bottom-Right shading for Top-Left and Bottom-Right and Improved Top-Left and Bottom-Right is shown in the Figure 10.

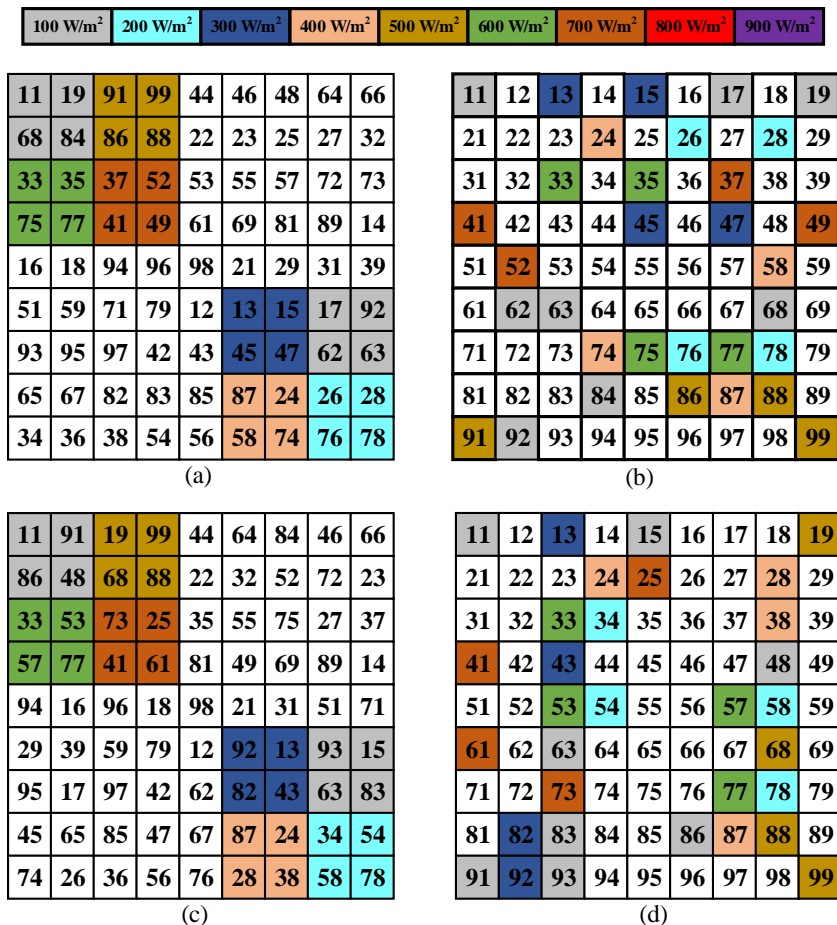

**Figure 10.** (**a**) OEP; (**b**) shade dispersion of OEP; (**c**) IOEP; (**d**) shade dispersion of IOEP for 9 × 9 PV array at Top-Left and Bottom-Right PSC.

The PV characteristics for case-V (Top-Right and Bottom-Right) are shown in Figure 11.

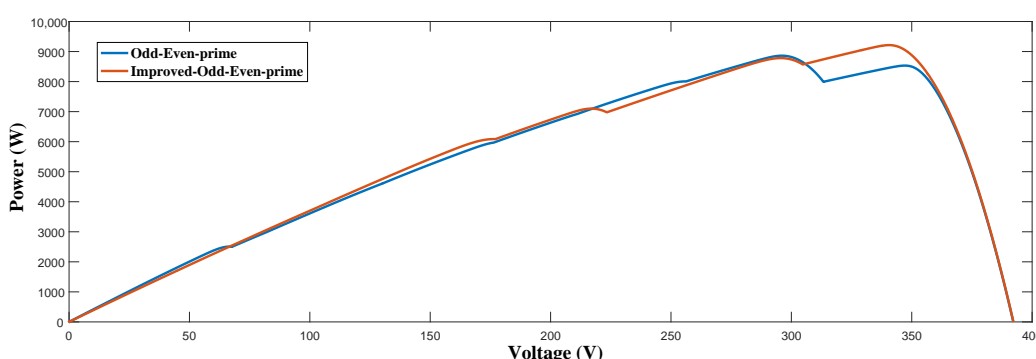

**Figure 11.** Voltage–power characteristics of 9 × 9 PV array for case-5 (Top-Left and Bottom-Right) PSC.

For different values of array current, the array voltage and power are calculated and noted in Table 6. From Table 6, it is clearly understood that the maximum value of power for OEP is **48** $V_m I_m$ and the maximum value of power for IOEP is **48.6** $V_m I_m$.

**Table 6.** GMPP for Group-V PSC.

| | Odd-Even-Primve | | | | Improved-Odd-Even-Primve | | |
| --- | --- | --- | --- | --- | --- | --- | --- |
| Row Bypassed | $I_a$ | $V_a$ | $P_a$ | Row Bypassed | $I_a$ | $V_a$ | $P_a$ |
| $I_{R9}$ | $4.9\ I_m$ | $9\ V_m$ | $44.1\ V_m I_m$ | $I_{R9}$ | $6\ I_m$ | $8\ V_m$ | $48\ V_m I_m$ |
| $I_{R8}$ | $6.8\ I_m$ | $5\ V_m$ | $34\ V_m I_m$ | $I_{R8}$ | $7.5\ I_m$ | $3\ V_m$ | $22.5\ V_m I_m$ |
| $I_{R7}$ | $7.9\ I_m$ | $2\ V_m$ | $15.8\ V_m I_m$ | $I_{R7}$ | $7.2\ I_m$ | $5\ V_m$ | $36\ V_m I_m$ |
| $I_{R6}$ | $7\ I_m$ | $4\ V_m$ | $28\ V_m I_m$ | $I_{R6}$ | $7.1\ I_m$ | $6\ V_m$ | $42.6\ V_m I_m$ |
| $I_{R5}$ | $8.1\ I_m$ | $1\ V_m$ | $8.1\ V_m I_m$ | $I_{R5}$ | $6.6\ I_m$ | $7\ V_m$ | $46.2\ V_m I_m$ |
| $I_{R4}$ | $6.3\ I_m$ | $7\ V_m$ | $44.1\ V_m I_m$ | $I_{R4}$ | $7.3\ I_m$ | $4\ V_m$ | $29.2\ V_m I_m$ |
| $I_{R3}$ | $6.0\ I_m$ | $8\ V_m$ | **$48\ V_m I_m$** | $I_{R3}$ | $7.5\ I_m$ | $3\ V_m$ | $22.5\ V_m I_m$ |
| $I_{R2}$ | $6.5\ I_m$ | $6\ V_m$ | $39\ V_m I_m$ | $I_{R2}$ | $5.4\ I_m$ | $9\ V_m$ | **$48.6\ V_m I_m$** |
| $I_{R1}$ | $7.1\ I_m$ | $3\ V_m$ | $21.3\ V_m I_m$ | $I_{R1}$ | $6\ I_m$ | $8\ V_m$ | $48 V_m I_m$ |

### 3.1.8. Performance Indices

The performance indices have been calculated, such as mismatch power loss, fill-factor, efficiency for OEP, and IOEP PV reconfiguration techniques.

A bar graph of mismatch power loss (MPL) is shown in Figure 12, where the maximum values for OEP and IOEP are 21.604% and 19.654%, respectively. This indicates that IOEP has a lower value of %MPL and more effectively distributes shade compared to the OEP configuration.

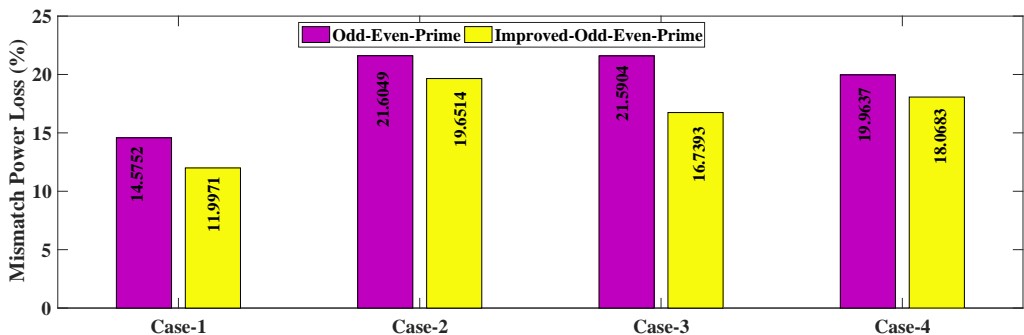

**Figure 12.** Mismatch power los s for $9 \times 9$ PV array.

Figure 13 presents the variation of fill-factor (%) for OEP and IOEP under various partial-shading conditions. The average values of Fill-Factor (%) for OEP and IOEP are 59.5850% and 61.6250%, respectively. The IOEP configuration has a higher average value of FF compared to the OEP configuration, indicating that it has superior performance.

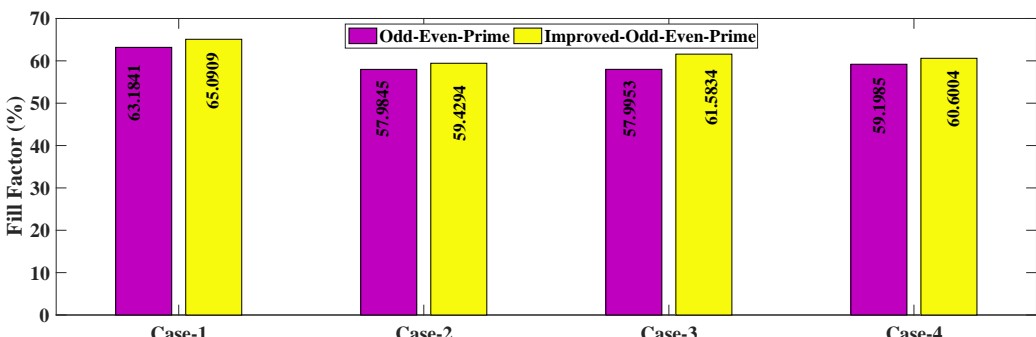

**Figure 13.** Fill-factor for $9 \times 9$ PV array.

Figure 14 shows a bar graph of efficiency. From the figure, it can be seen that the range of efficiency (%) for OEP and IOEP is 10.41 –11.34% and 10.67–11.68%, respectively. IOEP has the highest range of efficiency, indicating that it performs better than the OEP configuration.

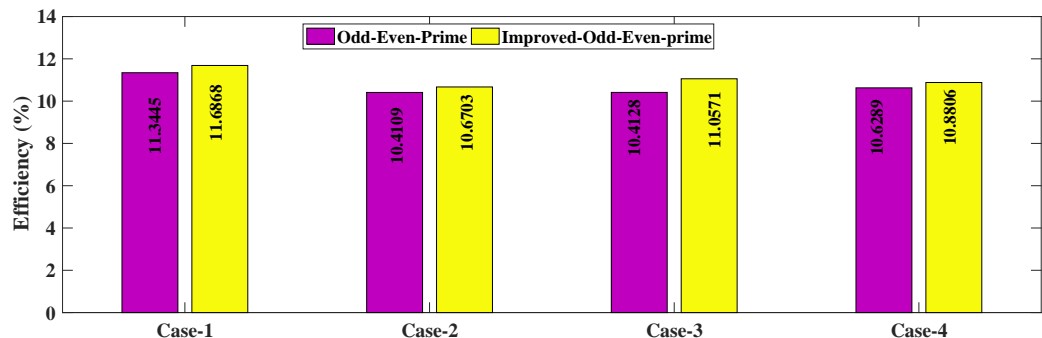

**Figure 14.** Efficiency for 9 × 9 PV array.

*3.2. Implementation of Improved OEP 8 × 9 PV array*

OEP and IOEP reconfigurations are mainly useful for the rectangular PV array. Therefore, the proposed IOEP reconfiguration tested on a 8 × 9 PV array. All the shadings have taken from the OEP.

Figure 15 shows the OEP and IOEP arrangement for 8 × 9 TCT configured PV array.

**Figure 15.** (**a**) OEP and (**b**) IOEP arrangement for 8 × 9 PV array.

### 3.2.1. Case-1 (Top-Left) Shading

In Case-1, the Top-Left corner, 16 PV modules are partially shaded, as seen in Figure 16a.

If the sum of the Irradiances of PV modules presented in each row is same then it indicate Uniform Irradiance condition . From Figure 16b, it is understood that in Row-4, four PV modules are subjected to PSC, whereas in Row-7, no PV module is subjected to PSC. From Figure 16d, it is understood that in Row-4, three PV modules are subjected to PSC, whereas in Row-7, one PV module subjected to PSC. From the above explanation it is understood that IOEP reconfiguration distributes the shade to all the rows effectively compared to OEP.

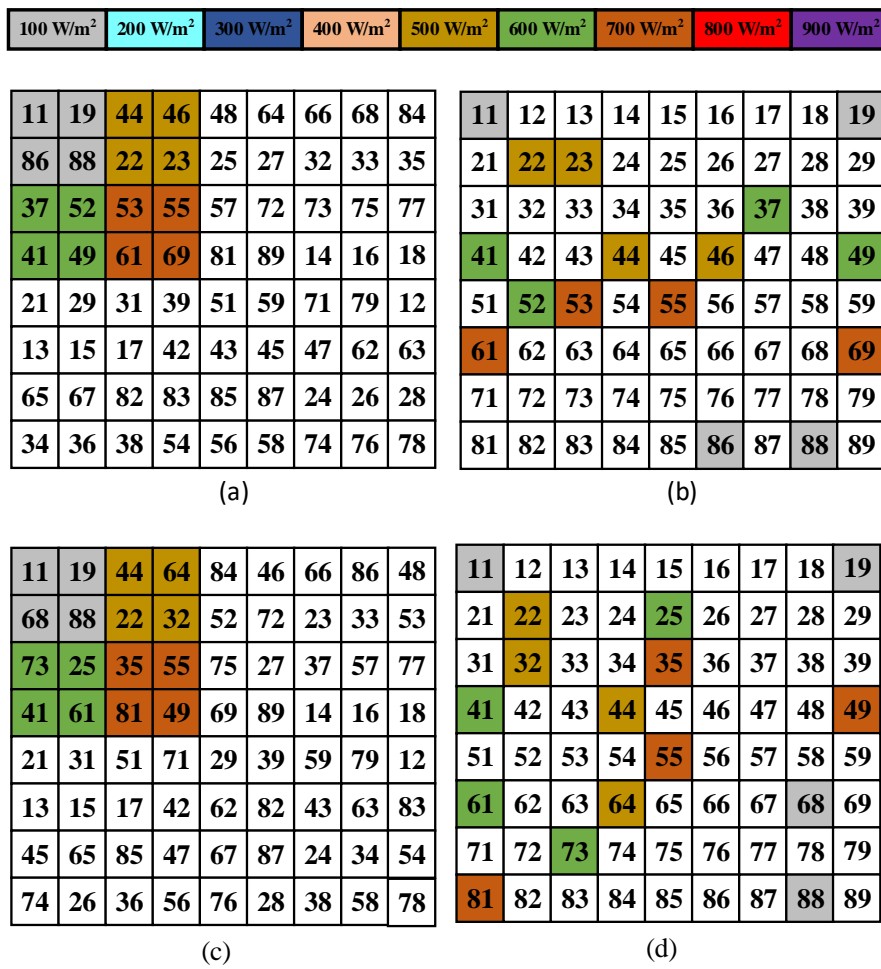

**Figure 16.** (**a**) OEP; (**b**) shade dispersion of OEP; (**c**) IOEP; (**d**) shade dispersion of IOEP for 8 × 9 PV array at Top-Left PSC.

The voltage–power characteristics for 8 × 9 PV array for case-1 (Top-Left) are shown in the Figure 17. From the voltage–power characteristics it is observed that IOEP reconfiguration generate more power compare to OEP reconfiguration under partial-shading conditions.

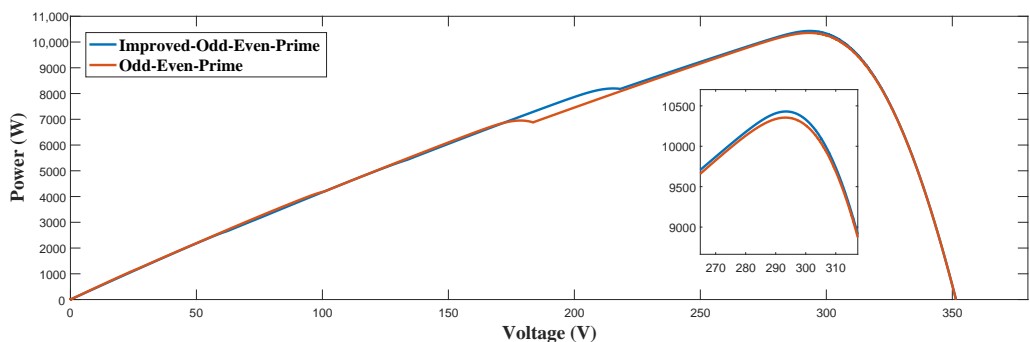

**Figure 17.** Voltage–power characteristics of 8 × 9 PV array for case-1 (Top-Left) PSC.

### 3.2.2. Case-2 (Top-Right) Shading

In Case-2, in the Top-Right corner, 16 PV modules are partially shaded, as seen in the Figure 18a.

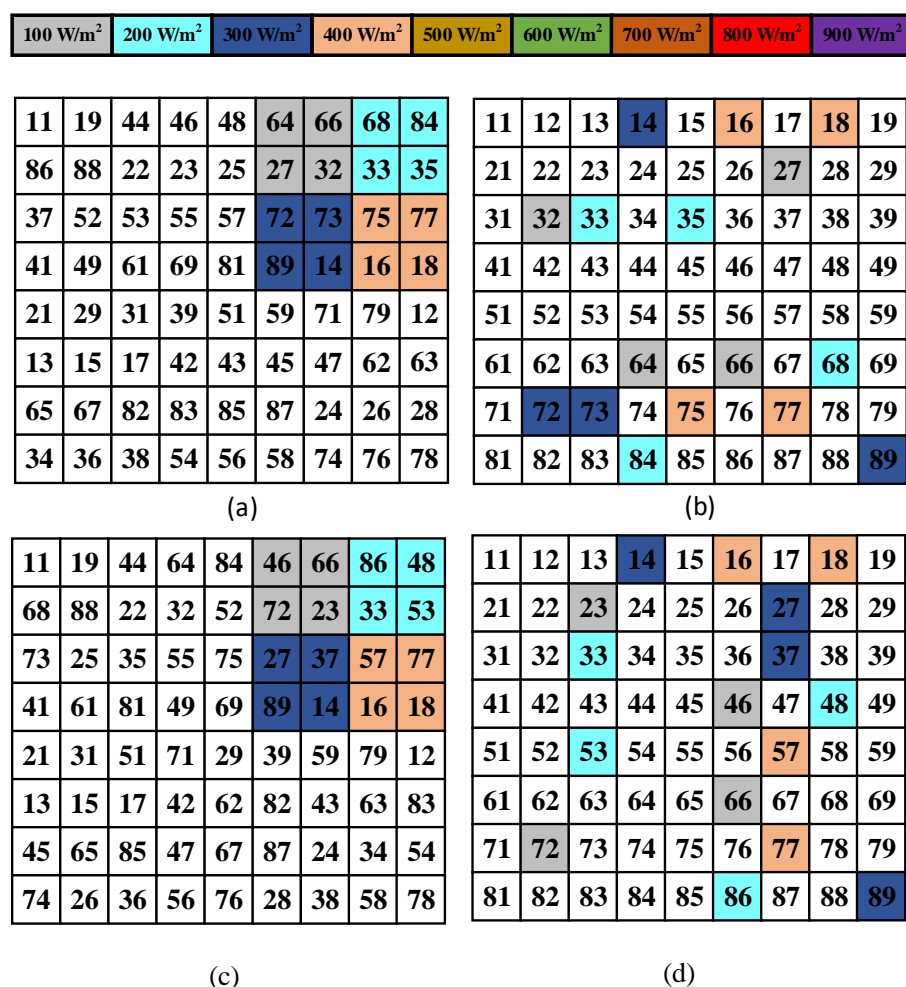

**Figure 18.** (**a**) OEP; (**b**) shade dispersion of OEP; (**c**) IOEP; and (**d**) shade dispersion of IOEP for 8 × 9 PV array at Top-Right PSC.

The voltage–power characteristics of case-2 (Top-Right) are shown in the Figure 19. From the voltage–power characteristics it understood that IOEP reconfiguration has a superior performance under partial-shading conditions.

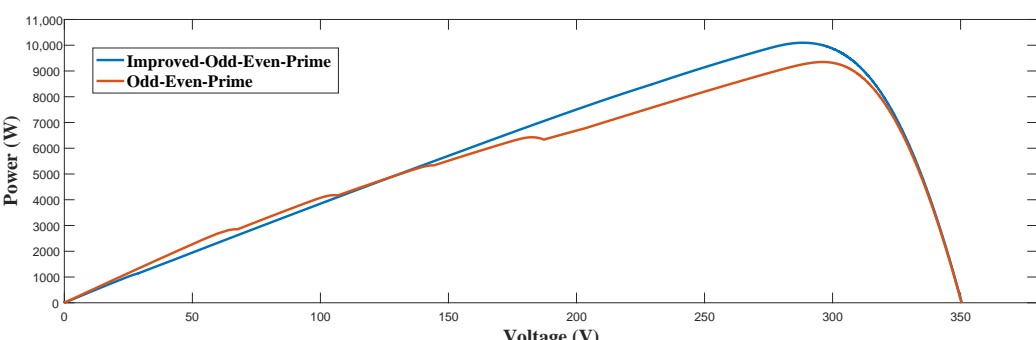

**Figure 19.** Voltage–power characteristics of 8 × 9 PV array for case-2 (Top-Right) PSC.

### 3.2.3. Case-3 (Bottom-Left) Shading

In Case-3, in the Bottom-Left corner, 16 PV modules are partially shaded, as seen in the Figure 20a.

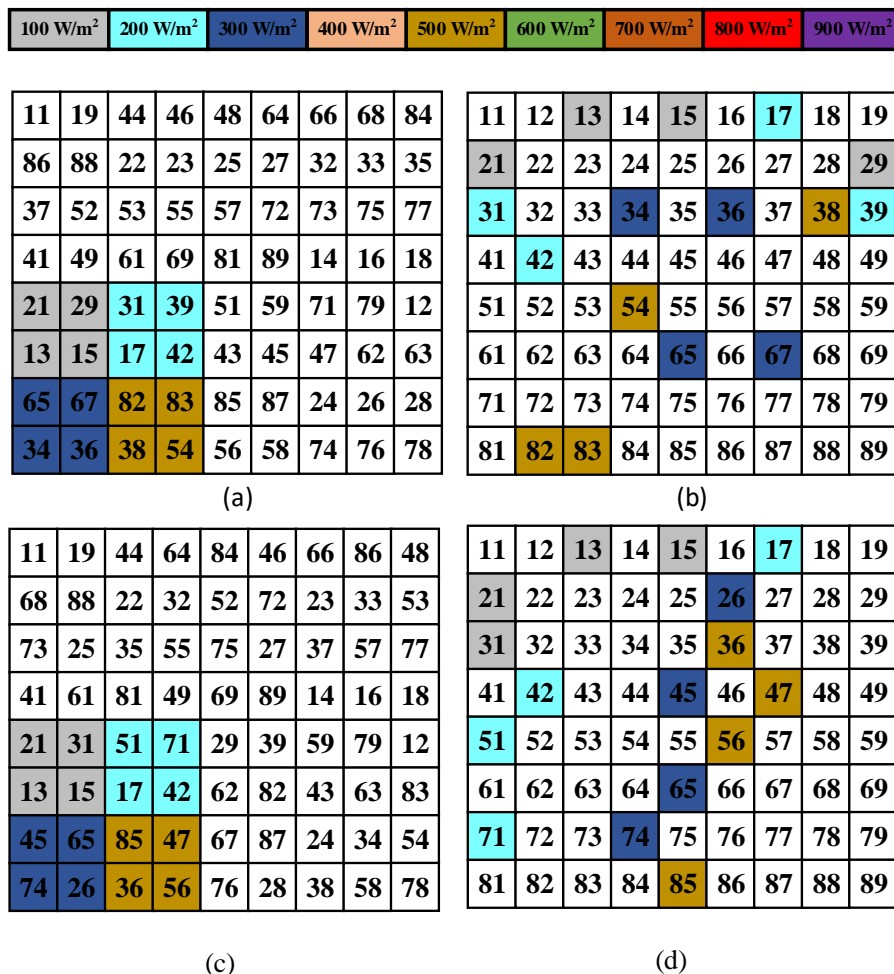

**Figure 20.** (**a**) OEP; (**b**) shade dispersion of OEP; (**c**) IOEP; and (**d**) shade dispersion of IOEP for Case-3 (Bottom-Left) PSC.

The voltage–power characteristics of case-3 (Bottom-Left) are shown in the Figure 21. From the voltage–power characteristics, it understood that IOEP reconfiguration has superior performance under partial-shading conditions.

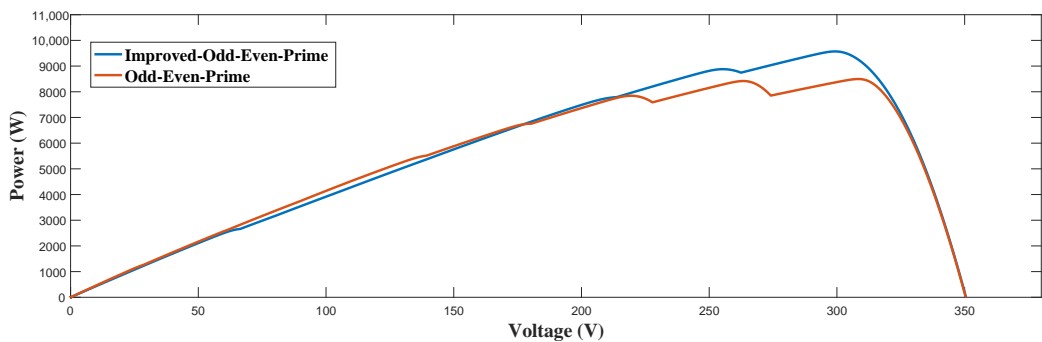

**Figure 21.** Voltage–power characteristics of $8 \times 9$ PV array at Bottom-Left PSC.

### 3.2.4. Case-IV (Bottom-Right) Shading

In Case-IV, in the Bottom-Right corner, 16 PV modules are partially shaded, as seen in the Figure 22a.

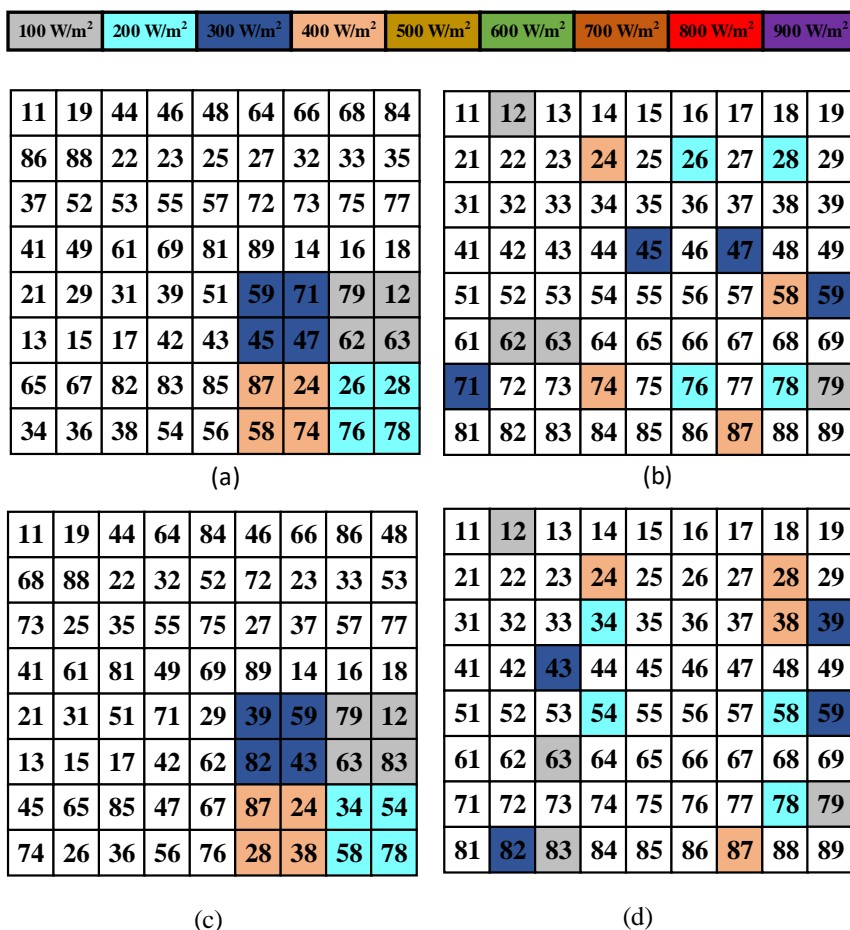

**Figure 22.** (**a**) OEP; (**b**) shade dispersion of OEP; (**c**) IOEP; and (**d**) shade dispersion of IOEP for 8 × 9 PV array at Bottom-Right PSC.

The voltage–power characteristics of case-IV (Bottom-Right) are shown in the Figure 23. From the voltage–power characteristics it understood that IOEP reconfiguration has superior performance under partial-shading conditions.

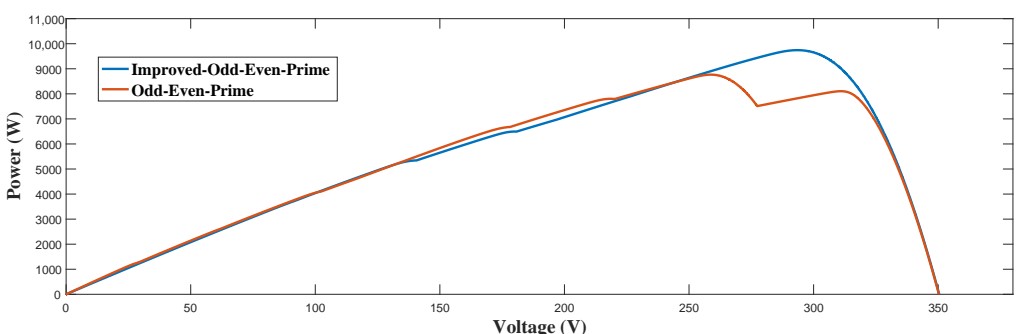

**Figure 23.** Voltage–power characteristics of 8 × 9 PV array for case-IV (Bottom-Right) PSC.

### 3.2.5. Case-5 (Top-Left and Bottom-Right) Shading

Case-5 is a combination of case-1 and case-4, i.e., in case-5 both Top-Left and Bottom-Right PSC have been considered. In case-5, a total of 32 PV modules were subjected to PSC, as shown in the Figure 24a.

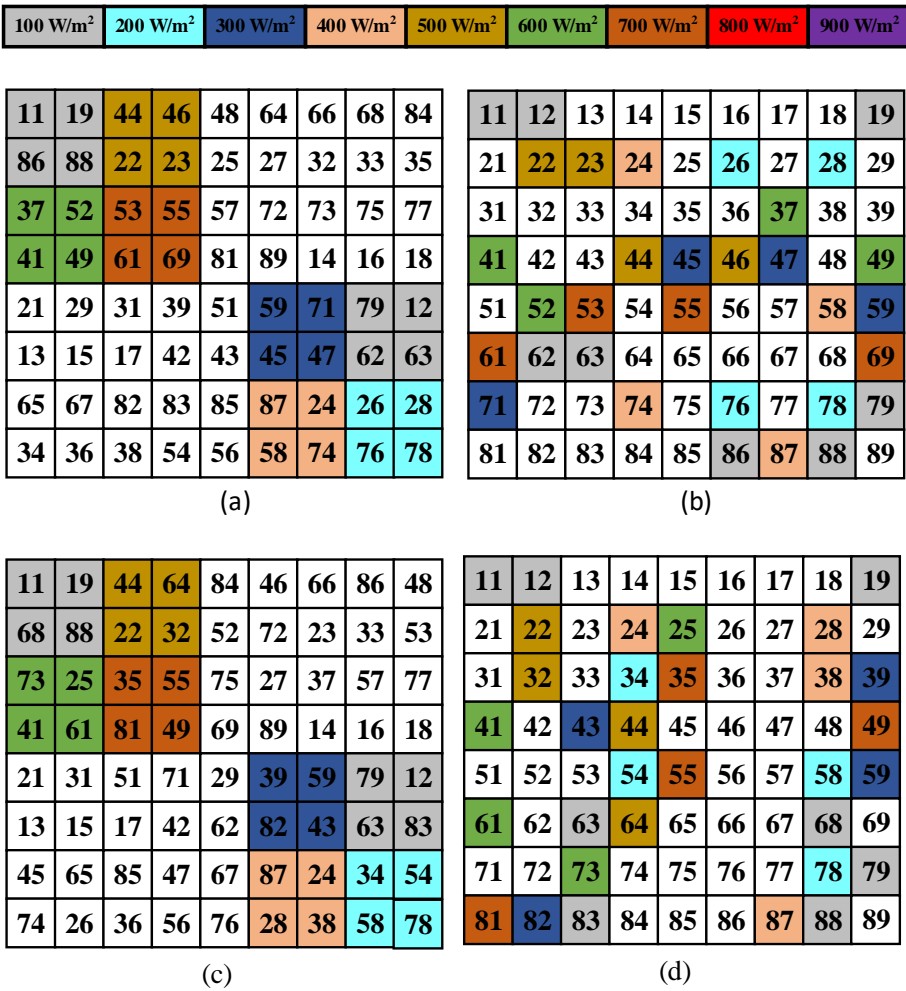

**Figure 24.** (**a**) OEP; (**b**) shade dispersion of OEP; (**c**) IOEP; and (**d**) shade dispersion of IOEP for 8 × 9 PV array at Top-Left and Bottom-Right PSC.

The voltage–power characteristics for case-5 (Top-Left and Bottom-Right) for 8 × 9 PV array is shown in the Figure 25. From the voltage–power characteristics, it is understood that IOEP reconfiguration has superior performance under partial-shading conditions.

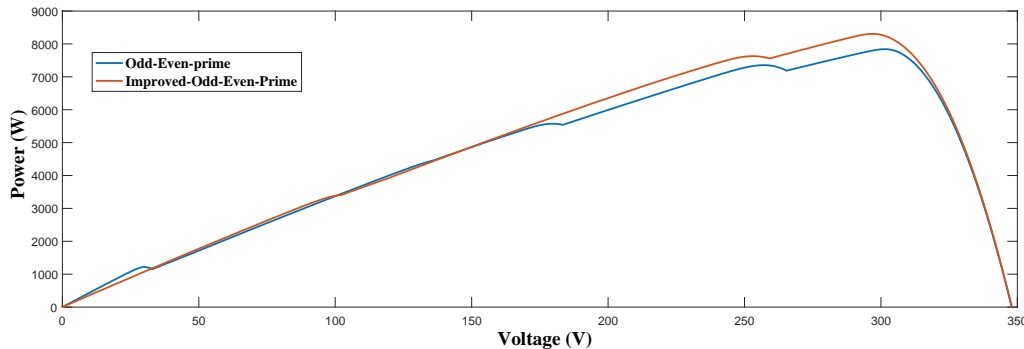

**Figure 25.** Voltage–power characteristics of 8 × 9 PV array for case-5 (Top-Left and Bottom-Right) PSC.

3.2.6. Performance Indices

The performance of OEP and IOEP PV reconfiguration techniques has been evaluated using indices, such as mismatch power loss, fill-factor, and efficiency. A bar graph of

mismatch power loss (MPL) is shown in Figure 26, with the maximum values for OEP and IOEP being 30.585% and 21.81%, respectively. This indicates that IOEP has a lower value of % MPL and more effectively distributes shade compared to the OEP configuration.

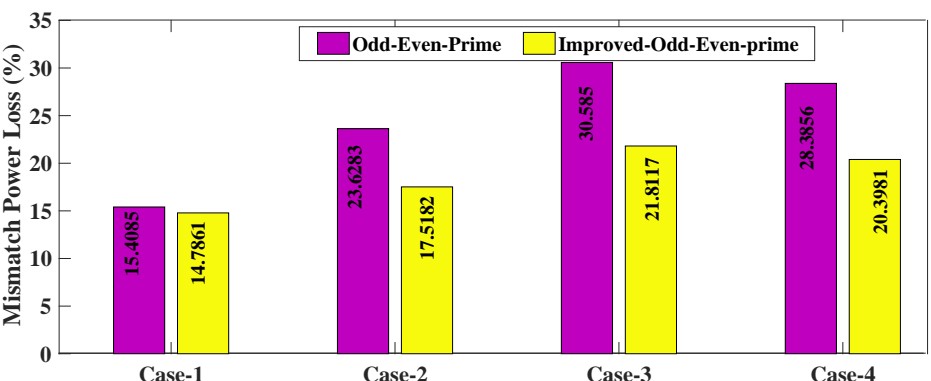

**Figure 26.** Mismatch power loss for 8 × 9 PV array.

Figure 27 illustrates the variation of fill-factor (%) for OEP and IOEP under different partial shading conditions. The average values of fill-factor (%) for OEP and IOEP are 56% and 60.82%, respectively. The IOEP configuration has a higher average value of FF compared to the OEP configuration, indicating that it has superior performance.

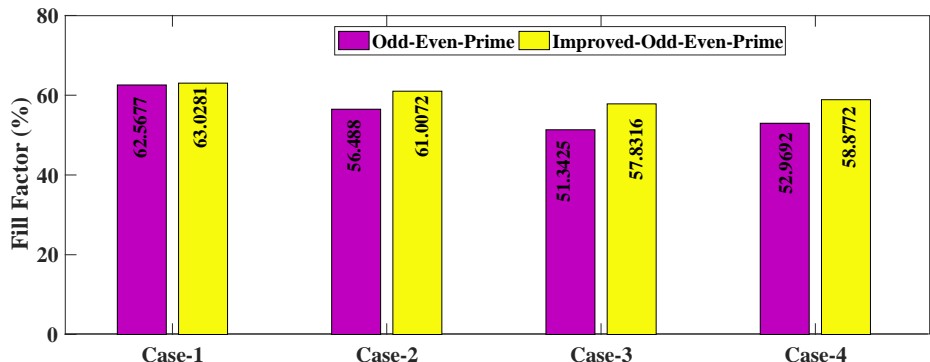

**Figure 27.** Fill-factor for 8 × 9 PV array.

A bar graph of efficiency is shown in Figure 28. From the graph, it can be seen that the range of efficiency (%) for OEP and IOEP is 10.14–11.23% and 10.38–11.41%, respectively. IOEP has the highest range of efficiency, indicating that it performs better than the OEP configuration.

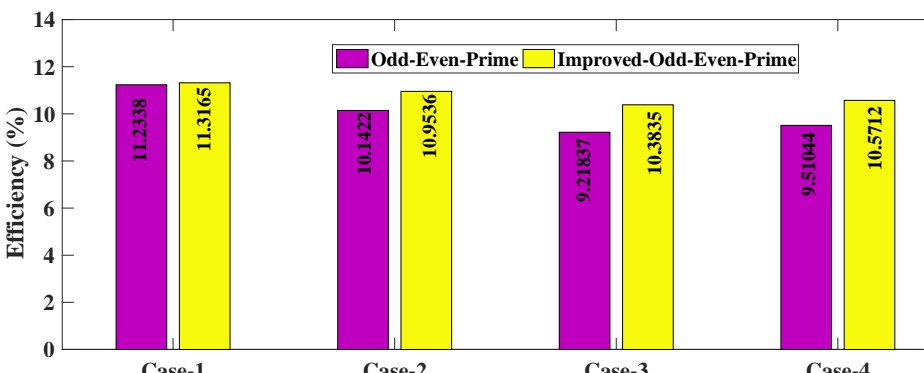

**Figure 28.** Efficiency for 8 × 9 PV array.

The GMPP for different shading patterns of OEP and IOEP reconfigurations are computed and noted in the Table 7. From Table 7, it is understood that IOEP reconfiguration has superior performance for all shading patterns.

**Table 7.** GMPP for $9 \times 9$ and $8 \times 9$ PV array for different shading patterns.

| Shading Pattern | 9×9 PV Array | | 8 × 9 PV Array | |
|---|---|---|---|---|
| | OEP GMPP(W) | IOEP GMPP(W) | OEP GMPP(W) | IOEP GMPP(W) |
| Case-1 (Top-Left) | 11,763 | 12,118 | 10,354 | 10,430 |
| Case-2 (Top-Right) | 10,795 | 11,064 | 9347 | 10,096 |
| Case-3 (Bottom-Left) | 10,797 | 11,465 | 8496 | 9570 |
| Case-4 (Bottom-Right) | 11,021 | 11,282 | 8765 | 9743 |
| Case-5 (Top-Left and Bottom-Right) | 8861 | 9213 | 7840 | 8306 |

## 4. Conclusions

In this paper, we introduce the Improved Odd–Even–Prime reconfiguration method as a way to increase the output power of a rectangular photovoltaic (PV) array under partial-shading conditions. We tested the performance of this method using various shading patterns and performance indices, such as fill-factor, efficiency, and mismatched power loss. The improved Odd–Even–Prime reconfiguration distributes the shading across all rows of the PV array, reducing the impact of shading on any individual row, and ultimately increasing the output power of the array. Our results demonstrate that the improved Odd–Even–Prime reconfiguration method leads to higher GMPP, fill-factor, and efficiency compared to the OEP PV reconfiguration technique.

**Author Contributions:** Conceptualization, D.K., C.Y. and S.R.S.; methodology, D.K. and C.Y.; software, D.K., C.Y. and S.R.S.; validation, C.Y., D.K. and S.R.S.; formal analysis, C.Y. and S.R.S.; investigation, D.K. and C.Y.; resources, C.Y. and S.R.S.; data curation, D.K. and C.Y.; writing—original draft preparation, D.K. and C.Y.; writing—review and editing, D.K., C.Y. and S.R.S.; visualization, C.Y. and S.R.S.; supervision, C.Y. and S.R.S.; project administration, C.Y. and S.R.S.; funding acquisition, C.Y. and S.R.S. All authors have read and agreed to the published version of the manuscript.

**Funding:** This research was funded by WOOSONG UNIVERSITY's Academic Research Funding—2023.

**Data Availability Statement:** Not applicable.

**Conflicts of Interest:** The authors declare no conflicts of interest.

## Abbreviations

**Acronyms**

| | |
|---|---|
| EAR | Electrical Array Reconfiguration |
| ONR | Odd-Numbered rows |
| E-E | Even–Even |
| ONC | Odd-Numbered columns |
| E-O | Even–Odd |
| ENR | Even-numbered rows |
| E-P | Even–Prime |
| ENC | Even-Numbered columns |
| FF | Fill-factor |
| GMPP | Global maximum power point |
| O-E | Odd–Even |

| | |
|---|---|
| P-E | Prime–Even |
| OEP | Odd–Even–Prime |
| PNR | Prime-numbered rows |
| IOEP | Improved Odd–Even–Prime |
| O-O | Odd–Odd |
| O-P | Odd–Prime |
| PNC | Prime-numbered columns |
| P-O | Prime–Odd |
| PSC | Partial-shading condition |
| PV | Photovoltaic |
| P-P | Prime–Odd |
| TCT | Total-Cross-Tied |

**Nomenclature**

| | |
|---|---|
| $m$ | Number of rows in PV array |
| $n$ | Number of columns in PV array |
| $M_{xy}$ | PV module location ($x^{th}$ row and $y^{th}$ column) before reconfiguration |
| $V_{oc}$ | Open-Circuit Voltage (V) |
| $I_{sc}$ | Short-Circuit Current (A) |
| $V_{mp}$ | Voltage at Maximum power point (V) |
| $I_{mp}$ | Current at Maximum power point (A) |
| $I_{row-m}$ | $m^{th}$ row current in TCT configuration (A) |
| $I_m$ | Maximum row current in TCT configuration (A) |
| $V_m$ | Maximum row voltage in TCT configuration (V) |

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
