# Peer review of "Improved-Odd-Even-Prime Reconfiguration to Enhance the Output Power of Rectangular Photovoltaic Array under Partial Shading Conditions"

_electronics, doi:10.3390/electronics12020427_

Round 1
Reviewer 1 Report
The review comments for the manuscript, 'Improved-Odd-Even-Prime Reconfiguration to enhance the Output Power of Rectangular Photovoltaic array Under Partial Shading Conditions', are given below,
1. The authors presented a new reconfiguration for partial shaded PV system. The work is impressive and presented well. Still, few changes need to be done for betterment.
2. Adding nomenclature for the symbols along with abbreviations used in the manuscript may help the readers.
3. Introduction section need to be improved a lot. It must be updated with more recent references; I suggest the following, 10.1109/ACCESS.2021.3138917, 10.1109/ACCESS.2021.3076608 and 10.1109/ACCESS.2022.3148065
4. Also add the organization of the manuscript at the Introduction section.
5. References should not be grouped like, ,[27–31] or [18–22]. Each and every reference must be properly justified and cited.
6. What type of cells were used for this work and there is no modelling part of PV array. Kindly refer an accurate analytical modeling of solar photovoltaic system considering Rs and Rsh under partial shaded condition, for modelling.
7. How the proposed method is better than the L-Shape Propagated Array Configuration With Dynamic Reconfiguration Algorithm? There is no comparison given with any other recent methods. It must be provided for validating the work.
8. English must be improved with the help of a language expert.
9. Typos and grammatical errors are throughout the manuscript. Kindly check it once.
Author Response
The authors would like to express their humble and sincere thanks to the academic editor and reviewers for spending their valuable time and providing various suggestions to improve the quality of the submitted paper. It is encouraging to see the appreciation of the reviewers through highlighting the merits of the paper. Besides, the comments and the modifications suggested by them are meaningful and helped a lot to make the manuscript better.
Please see the attachment with the response to the comments. Thank you.

Reviewer 2 Report
The study presented in this research is sound, and the results produced are interesting. But a
revision is required, and after responding to the following remarks and revising the paper, the manuscript may be considered for publication.
1. Literature review needs to include several recent, relevant publications (high impact) highlighting their key findings. The current version only discussed general aspects while the review of each from several papers is necessary.
2. Authors are suggested to include a separate paragraph discussing the novelty and importance of the present work.
3. Improve the quality of figures 3,5,7 and 9 (make the axis clear).
4. It would be better if you run a fifth case (mix case no 1 with case no 4) to have ashading pattern Top-left with bottom right.
5. Also, check the typos throughout the manuscript during revision submission (for examble line 8 in the abstract).
Author Response

(The authors gave the same response as above.)
